# Osteosarcopenia: A Narrative Review on Clinical Studies

**DOI:** 10.3390/ijms23105591

**Published:** 2022-05-17

**Authors:** Angela Polito, Lorenzo Barnaba, Donatella Ciarapica, Elena Azzini

**Affiliations:** CREA-Research Centre for Food and Nutrition, Via Ardeatina 546, 00178 Rome, Italy; lorenzo.barnaba@crea.gov.it (L.B.); donatella.ciarapica@crea.gov.it (D.C.)

**Keywords:** osteosarcopenia, clinical studies, interventions

## Abstract

Osteosarcopenia (OS) is defined by the concurrent presence of osteopenia/osteoporosis and sarcopenia. The pathogenesis and etiology of OS involve genetic, biochemical, mechanical, and lifestyle factors. Moreover, an inadequate nutritional status, such as low intake of protein, vitamin D, and calcium, and a reduction in physical activity are key risk factors for OS. This review aims to increase knowledge about diagnosis, incidence, etiology, and treatment of OS through clinical studies that treat OS as a single disease. Clinical studies show the relationship between OS and the risk of frailty, falls, and fractures and some association with Non-communicable diseases (NCDs) pathologies such as diabetes, obesity, and cardiovascular disease. In some cases, the importance of deepening the related mechanisms is emphasized. Physical exercise with adequate nutrition and nutritional supplementations such as proteins, Vitamin D, or calcium, represent a significant strategy for breaking OS. In addition, pharmacological interventions may confer benefits on muscle and bone health. Both non-pharmacological and pharmacological interventions require additional randomized controlled trials (RCT) in humans to deepen the synergistic effect of exercise, nutritional interventions, and drug compounds in osteosarcopenia.

## 1. Introduction

The term “osteosarcopenia” (OS) has been recently proposed to define the concurrent presence of osteopenia/osteoporosis and sarcopenia [1,2,3,4] Osteoporosis and osteopenia are characterized by different grades of low bone mass and deterioration of bone tissue and are associated with an increase in bone fragility. Sarcopenia is a syndrome characterized by progressive and generalized loss of skeletal muscle mass and strength associated with the risk of adverse outcomes such as physical disability, poor quality of life, and death [5].

The pathogenesis and etiology of OS with the involvement of several factors including genetic, biochemical, mechanical, and lifestyle, represents a new emerging public issue, especially in older health. The co-existence of the low mass of bone and its micro-architectural deterioration (osteoporosis), and the loss of muscle mass, strength, and function (sarcopenia) have been highlighted as worsening outcomes than each one alone [6,7]. Musculoskeletal health depends on the close relationship between bone mass and muscle mass involving mainly two stages of the life course, i.e., early during development and growth in which children/adolescents build up their biological reserves which influence the later stage when there is an increased risk of muscle wasting and osteoporosis [8]. As shown in Figure 1, the identified muscle-bone unit [9] interacts through multiple communication pathways, whereby muscle receives signals from bone and vice versa. First, bones and muscles along with joints, cartilage, tendons, ligaments, and connective tissue make up the musculoskeletal system, so they are strictly connected. This contributes to the understanding that their actions are rarely independent. These tissues are highly vascularized providing a possible route in skeletal muscle that allows the secretion of molecules into circulation through the extracellular fluid present in the endomysium, and, due to their proximity, myokines and small molecules can cross the periosteum by diffusion [10]. In bone, osteocytes and their dendritic extensions are in canaliculi forming a remarkable, highly branched, lacunar-canalicular signaling network close to the blood vessels easing the transport of osteokines, growth factors, hormones, and other bioactive molecules [11]. Second, bone, fat, and skeletal muscle originate from the same progenitor cells, the mesenchymal stem cells (MSCs), which are multipotent stromal cells able to generate cells with different phenotypes including osteoblasts (bone), adipocytes (fat), or chondrocytes (cartilage). Adult stem cells and MSCs reside within multiple connective tissue depots and keep the ability to self-renew and repair/regenerate aging and damaged tissues and exist in complex paracrine and endocrine feedback loops among bone, fat, and skeletal muscle mediated by chemicals, cytokines, hormones, and other molecules [12]. Adipogenesis has been defined as a consequence of biological aging by the MSCs differentiation pathway [13]. Critical steps at the onset of OS can be represented by the presence of marrow adiposity and/or intramuscular adipose tissue. The presence of adipocytes could affect the microenvironment altering myogenesis and osteogenesis producing localized adipokines, free fatty acids, and lipids and so inducing local lipotoxicity and decreasing bone formation and increasing bone resorption [14]. Correspondingly, fat infiltration in muscle fibers is associated with cell dysfunction [15]. In the muscle, fat can be located between muscle bundles as a depot of adipocytes (intramuscular fat) and/or within myofibers lipid as infiltration (intramyocellular fat). These infiltrations, due to several mechanisms, including de-differentiation of muscle-derived stem cells or other mesenchymal progenitors into cells with adipocyte phenotype, seem to be associated with insulin insensitivity, inflammation, and functional deficits in skeletal muscle [16,17]. Another crucial pathway for keeping bone mass and strength as well as the regulation of bone remodeling is the growth hormone (GH)-insulin-like growth factor 1 (IGF-1) axis, (GH-IGF-1 axis). GH is a single-chain polypeptide hormone that is secreted in a pulsatile manner by the anterior pituitary gland in response to hypothalamic stimulation, and its deficiency causes a reduction in muscle and bone mass and an increase in fat mass. Lindsay and Mohan [18] proposed a model of GH/IGF-I regulation of skeletal growth, including both endocrine and local actions of IGF-I. GH acts by increasing hepatic IGF-I production, or by influencing the bone directly, independent of IGF-I. Hepatic IGF-I acts in an endocrine manner by circulating in the blood, or locally, and its production by specific tissues, including the bone and muscle, acts in autocrine and paracrine signaling. Lastly, particularly during aging, physical inactivity and a positive energy balance may favor a chronic low-grade inflammation resulting in a shift of mesenchymal stem cell lineage towards adipogenesis and away from myogenesis and osteoblastogenesis. Fatty infiltration into muscle and bone tissues and progressive replacement of muscle and bone cells with fat cells leads to the recruitment of immunological cells with the release of pro-inflammatory cytokines causing chronic low-grade inflammation [19]. Moreover, an inadequate nutritional status, such as low intake of protein, vitamin D, and calcium, and a reduction in physical activity are key risk factors for OS. 

Preventing and treating muscle and bone mass loss, especially in the elderly population who are at higher risk, could contribute to the prevention of sarcopenia and osteopenia/osteoporosis. This goal would decrease the risk for falls and fractures, making older individuals less susceptible to the development of mobility limitations or severe disabilities that ultimately affect their capacity for independence [20]. Although a decrease in muscle function could relate to a decrease in bone mass due to a decline in mechanical loading on the bone [21], a plethora of scientific studies considered independently the two syndromes as risk factors for falls, fracture, and mortality, especially in older people. 

This narrative review aims to increase knowledge about the diagnosis, incidence, etiology, and treatment of OS considering only clinical studies that treat OS as a single disease. Relevant studies were searched through the electronic databases PubMed, Scopus, and Google Scholar from the start to January 2022. The search strategy used included the keywords “osteosarcopenia” and “clinical trial” or “clinical study” or “randomized controlled trial” in humans. No geographical restrictions were applied but the language had to be English. Unpublished studies/grey literature were not considered in our work.

## 2. Diagnostic Criteria

OS does not have a unique model of diagnosis but is based on the reference definitions of osteoporosis and sarcopenia and some studies point to the need to standardize these definitions to improve the diagnosis of OS [22,23,24]. The European Working Group on Sarcopenia in the Elderly (EWGSOP2) met in 2018 to update the original definition of sarcopenia following the scientific and clinical evidence developed over the past decade. The operational definition of sarcopenia is based on 1. Low muscle strength; 2. Low muscle quantity or quality; 3. Low physical performance. In particular, low muscle strength with a grip strength of <27 kg for men and <16 kg for women, or the time for a chair stand test >15 s for 5 rises, and low muscle quantity or quality determined by height-adjusted appendicular lean mass (ALM/height2) less than 6.0 kg/m^2^ in women and less than 7.0 kg/m^2^ in men or only an ALM of 20 kg for men and 15 kg for women are the cut-offs used [25]. If a diagnosis of sarcopenia is present, a low bone mass determined by the T-score Bone Minela Density (BMD—lumbar spine, femoral neck, and total femur) < –1.0 SD (osteopenia/osteoporosis) or a T score ≤ –2.5 SD (osteoporosis) [26] confirms a diagnosis of OS. Magnetic resonance imaging (MRI), computed tomography (CT), and dual-energy X-ray absorptiometry (DXA) are sophisticated techniques to assess both muscle and bone tissue composition. Some authors suggest also using methods more feasible for routine clinical uses as a means of diagnosis for OS such as bioimpedance analysis (BIA) that showed strong agreement with DEXA [27]. Others suggest the use of handgrip strength, pointing out that each 1-unit decrease in handgrip strength increased the risk of OS by 1.162 times [28]. 

Additionally, the biochemical assessment of bone and muscle metabolism has been proposed to improve early diagnosis and screening, and assess the response to therapies in people with OS. Fathi and colleagues [29] investigated the association of bone turnover markers such as osteocalcin (OC), C-terminal cross-linked telopeptide (CTX), tartrate-resistant acid phosphatase (TRAP), alkaline phosphatase bone (BALP), and other factors such as vitamin D, calcium, phosphorus, and alkaline phosphatase with OS in 400 elderly people. The study results showed a statistically significant difference in OC, CTX, and TRAP between osteosarcopenia (−) and osteosarcopenia (+) people. No statistically significant differences were seen in BALP, vitamin D, calcium, and phosphorus between the compared groups. OC and CTX were associated with an increased likelihood of OS (adjusted OR = 1.023 per OC; adjusted OR = 4.363 for CTX). The authors concluded that a valuable association between a high level of OC and CTX and the incidence of OS, which these bone turnover markers can easily check in serum sampling, improve the early diagnosis, screening, and assessment of the response to therapies in people with OS, as well as play an important role in estimating the risk incidence. Poggiogalle and colleagues [30] assessed the prevalence of body composition phenotypes (sarcopenia, osteoporosis, and their overlaps) in nonagenarians, and examined their relationships with IGF-1 status and physical functionality. The study was conducted on 87 subjects (37 men and 50 women). The results showed in osteosarcopenic men IGF1-Standard Deviation Score (SDS) values (−0.61 ± 0.37 vs. −0.04 ± 0.52, P.02) were lower than those in control group males, while IGF1-SDS were similar in women. Moreover, the men’s appendicular lean mass (ALM) was positively associated with IGF1-SDS values (*p* = 0.01) independent of age and C-reactive protein concentration while no association between IGF1-SDS values and BMD was found. Finally, IGF1-SDS was not associated with functional performance (CS-PFP) in men and women. The authors concluded that IGF1 sensitivity in skeletal muscle and bone may differ by sex in the elderly and IGF1 status did not appear to affect physical functionality, so it is important to find determinants and characteristics of OS to define conclusive diagnostic criteria. Circulating osteoprogenitor (COP) cells are blood-borne cells that express a variety of osteoblastic markers [31]. A growing body of evidence supports their role in various physiological and pathological processes, ranging from pubertal growth spurts and fracture repair [31] to disability and frailty in older subjects [32]. COP cell levels have been associated with postmenopausal-osteoporotic bone where a low BMD in postmenopausal women has an inverse relationship with COP cell levels [33]. However, knowledge of the role of COP cells in normal physiology is still limited. Gunawardene and colleagues [34] investigated the role of COPs in 144 healthy volunteers by trying to find a reference range of COP cells, measured by flow cytometry, as well as any potential changes related to age and gender. The authors have found a normal reference range of % COP that could be used in future studies looking at applications of COP cell quantification and/or isolation in clinical practice, in particular, the diagnosis and therapeutics of OS and other age-related musculoskeletal diseases. COP cells have been correlated with disability, frailty, and poor physical performance in older people where low percentages of COP cells may increase the risk of such diseases. However, Al Saedi and colleagues [35] proposed a more accurate indicator for frailty, namely the expression of lamina A, as studies examining age-related changes in % COP cells have shown divergent results. Furthermore, low levels of expression of lamin A are associated with OS in mice. COP cells and buccal swabs were obtained from 66 subjects with an average age of 74-years as part of the Nepean Osteoporosis and Frailty (NOF) Study. The authors assessed physical performance, disability, and frailty, and lamin A expression in epithelial and COP cells was quantified by flow cytometry. The results showed frail individuals have 60% lower levels of lamin A compared to non-frail individuals (*p* < 0.001) and 62% lower levels compared to pre-frail persons (*p* < 0.001), highlighting that lamin A expression in COP cells is a strong indicator of frailty, as already observed in mice, while further longitudinal studies are needed for confirming its validity as a diagnosis mark of OS. Studies on COP cells are recent and further research is needed to find the signaling pathways and mechanisms of action associated with various pathological conditions, including osteoporosis, fractures, vascular calcifications, fragility, and OS, as well as the effect of supplementation and diet on these cells.

## 3. Clinical Studies on Osteosarcopenia and Associated Risk Factors

Sarcopenia and osteoporosis are diseases that have negative consequences and participate in physical decline, particularly in the elderly. Studies on OS and the co-presence of both diseases have shown an additive and synergistic effect that may contribute to a worsening of health outcomes, including the risk of falls, frailty, and death [36,37,38]. Huo and colleagues [39] in a cross-sectional study of 700 subjects showed that osteosarcopenic patients were older, mostly women, with a body mass index (BMI) below 25, and at higher risk of depression and malnutrition. The latter was also highlighted by Reiss and colleagues [40] and Okayama and colleagues [41] who add low quality of life as a risk factor for OS. Park [42] found that osteosarcopenic patients have greater frailty and disabilities in everyday life and worse depression. The results of another study [43] reinforced the above risk factors with osteosarcopenic patients preesnting as older and frailer with a lower BMI, fat and muscle mass, handgrip strength, and T-score compared to non-osteosarcopenic patients. Table 1 summarizes the relevant results of the main clinical studies on OS.

### 3.1. Osteosarcopenia and the Association with Falls, Fractures, and Frailty

Salech and colleagues [44] recruited more than 1100 subjects with a mean age of 72 years. OS was present in 16.4% of the sample and was associated with an increased risk of falls, fractures, and mortality. The prevalence of OS increased with age, reaching 33.7% for those older than 80 years and the mortality was significantly higher for the group with OS (15.9%) compared with those without the condition (6.1%). Additionally, the risk of falls, fractures and functional impairment was more frequent among osteosarcopenic patients than in those without the condition (falls: HR 1.60; CI 1.07–2.38; *p* < 0.05; fractures HR 1.54; CI 1.13–2.08; *p* < 0.01; functional impairment: HR 1.83; CI 1.41–2.38; *p* < 0.001). Additionally, Fahimfar and colleagues [45] identified a relationship between OS and falls. The authors showed that the risk of falls in osteosarcopenic subjects is positively associated with age (OR = 1.09, 95%CI: 1.04–1.14) and fasting blood glucose, and an increase of 10 mg/dL increased the chance of falling by 14% (OR = 1.14, 95% CI = 1.06–1.23), while it is negatively associated with systolic blood pressure and triglyceridemia (OR = 0.33, CI 95% = 0.12 to 0.89). Other clinical studies have investigated the relationship between OS and fractures. Di Monaco and colleagues [46] concluded that subjects with low bone mass and low muscle mass have an increased risk of vertebral fractures. Furthermore, the concomitant presence of sarcopenia and osteoporosis was associated with a higher Spine Deformity Index (SDI) score than the presence of only one of the two conditions in a sample of 350 women with subacute hip fracture. Turkmen and Ozcan [47] evaluated the relationship between BMD, gluteus maximus muscle volume (GMV), and hip fracture type in 134 patients and suggested measuring total gluteus maximus volume (GMV) as a tool to assess fracture risk in addition to examining BMD. OS can affect other physical performance as well as increase the risk of falls and frailty. Drey and colleagues [48] investigated the physical performance and bone metabolism in community-dwelling older adults. Sixty-eight prefrail adults between 65 and 94 years of age were recruited and assigned to four groups based on mean dual-energy X-ray absorptiometry results: osteosarcopenic, sarcopenic, osteopenic, and control. Physical performance was measured using the Short Physical Performance Battery (SPPB) score. The results showed a significantly reduced handgrip strength, increased chair lift time, and STS (sit-to-stand test) power time, as well as a significant increase in bone turnover markers only in osteosarcopenic patients. 

OS can also increase the likelihood that an individual will become frail. In a clinical study of 291 patients with chronic liver disease, including 137 males and 154 females with an average age of 70 years, the OS group showed a significantly higher prevalence of frailty (79.6% vs. 17.4%) and vertebral fracture (59.2% vs. 20.2%) than the non-OS group (*p* < 0.001) [49]. These results show that OS and fragility are closely related, and co-presence increases susceptibility to vertebral fractures and leads to impaired physical function. Inoue and colleagues investigated the relationship between OS and social [50] and cognitive frailty [51]. Social frailty is a common condition among older people and results in a loss of independence in activities of daily living and reflects loneliness, economic burden, and reduced social participation. While cognitive frailty, defined as a condition of reduced cognitive reserve, is considered to coexist with mild cognitive impairment (MCI) and physical frailty [52]. In 495 elderly patients [50] (mean age = 76.5 ± 7.2 years) classified as robust (58.2%), with osteoporosis alone (17.2%), with sarcopenia alone (13.5%), and with OS (11.1%), the authors found that social frailty prevalence increased stepwise from 8.0% in robust patients to 11.8%, 17.9%, and 29.1% among those with osteoporosis alone, sarcopenia alone, and OS, respectively (*p* < 0.001). The authors concluded that further studies are needed to clarify the causal relationship between OS and social frailty for improved health and longevity and for the prevention of disability in older adults. In a subsample of 432 patients [51], the prevalence of cognitive frailty among all patients was 20.8% and increased stepwise in the order of robust (14%), osteoporosis alone (22.4%), sarcopenia alone (34.7%), and OS (45.5%) (*p*< 0.001). Various mechanisms have been hypothesized to explain the findings, for example, it has been recognized that estrogen was related to cognitive functions, especially memory, and bone formation. Therefore, the association between OS and disorientation may reflect an estrogen-affected relationship between osteoporosis and cognitive function, especially memory function. However, these relationships are unclear and require further investigation.

### 3.2. Osteosarcopenia and Type 2 Diabetes Mellitus

Type 2 diabetes mellitus (T2DM) is a risk factor for OS and the co-occurrence of both may worsen musculoskeletal health and increase the risk of falls, fractures, and physical frailty [53]. β-cell function may be implicated in this relationship, which is positively associated with skeletal muscle mass index but not with body mineral density. Liu and colleagues [54] investigated the association between OS and β-cell function, as well as insulin resistance in patients with T2DM in a sample of 150 non-obese subjects aged ≥50 years. The results show that diabetic patients with OS have lower BMI, waist circumference, body fat percentage, and worse β-cell function. Furthermore, the relationship was independent of other OS risk factors, such as age, duration of diabetes, smoking, drinking, malnutrition, and glucose control. The authors conclude that β-cell function could be a protective factor against OS and preserving β-cell function is a preventive action toward OS in patients with T2DM. Pechmann and colleagues [55] evaluated the prevalence of OS and the association with trabecular bone score (TBS) in a group of patients with type 2 diabetes mellitus (*n* = 177, 64.4% women) and a control group (*n* = 146, 54.7% women), mean age 65.1 ± 8.2 years and 68.8 ± 11.0 years, respectively. The results show in diabetic patients a higher rate of OS (11.9% vs. 2.14%, respectively, *p* = 0.010), fractures (29.9% vs. 18.5%, respectively, *p* = 0.019), and lower handgrip strength values (24.4 ± 10.3 kg vs. 30.9 ± 9.15 kg, respectively, *p* < 0.001), but comparable BMD values. The authors conclude that the presence of OS in patients with type 2 diabetes mellitus was associated with chronic complications of diabetes, but not with increased fasting glucose levels or glycated hemoglobin (HbA1c). Thus, diabetic patients have a higher prevalence of OS and degraded TBS, and OS is associated with complications of diabetes, but not with diabetes duration or glycaemic control.

### 3.3. Osteosarcopenia and Obesity

Muscle, bone, and fat are closely related and changes in body composition can affect the whole body. An excess of body fat in a patient suffering from both sarcopenia and osteoporosis or osteopenia results in “Osteosarcopenic Obesity”. In a study of 1344 postmenopausal women over the age of fifty, the prevalence of osteosarcopenic obesity was 32% [56]. In addition, the dietary inflammatory index, used to measure the inflammatory degree through the diet, was evaluated. The results show an increased risk of osteosarcopenic obesity with a pro-inflammatory diet, in particular, a deficiency of antioxidant vitamins (A and E) was found in osteosarcopenic subjects compared to control (*p* < 0.001). This study can improve knowledge on the management of osteosarcopenic patients and further studies are needed on the interaction between muscle, bone, and fat, as well as in the prevention and treatment of related diseases. In addition to suffering from a worse inflammatory state, obese osteosarcopenic patients showed significantly lower levels of vitamin D and high parathyroid hormone (PTH) with normal renal function [57]. This combination of low vitamin D and high PTH is an important risk factor for falls and fractures so adequate vitamin D supplementation is recommended. Furthermore, OS is not limited to the elderly, but can also be found in younger persons. The study was conducted in a healthy lean group that included 1072 participants, and a healthy overweight/obese group that included 1479 participants [58]. The authors used an advanced bio-impedance device to assess the body composition and measured circulating high-sensitivity C-reactive protein (hsCRP) and diurnal salivary cortisol concentrations, as indices of inflammation and chronic stress. The results revealed a significant change in body composition in the young overweight/obese group, like the osteosarcopenic obesity seen in the middle-aged and the elderly populations. These data may be a ‘precursor’ of the osteosarcopenic obesity phenotype in young healthy overweight/obese subjects, who may progressively develop OS in its full form at an older age.

### 3.4. Osteosarcopenia and Polycystic Ovary Syndrome

Polycystic ovary syndrome (PCOS) is a complex endocrine disorder and carries significant health risks which are features of aging including an increased likelihood of impaired musculoskeletal health [59,60]. The obesity, insulin resistance (IR), sex hormone abnormalities, chronic inflammation, altered vitamin D status, and sedentary lifestyle seen in PCOS [61] are consistent with the pathophysiological mechanisms of OS. Kazemi and colleagues [62] investigated the relationship between OS and PCOS. The authors evaluated skeletal muscle index and BMD in 203 women aged 18–48 years who showed no symptoms of the menopausal transition. The results show a decrease in skeletal muscle index (SMI) % (mean [95% confidence interval (CI)]; 26.2% [25.1, 27.3] vs. 28.8% [27.7, 29.8]), lower limb SMI% (57.6% [56.7, 60.0] vs. 62.5% [60.3, 64.6]) and BMD (1.11 [1.08, 1.14] vs. 1.17 [1.14, 1.20] g/cm^2^) in the group with PCOS compared to controls. In addition, the PCOS group had elevated fasting insulin levels, Homeostatic Model Assessment for Insulin Resistance (HOMA-IR), and lower sex hormone-binding globulin (SHBG) and Matsuda index levels (to measure insulin sensitivity) than controls (*p* ≤ 0.04). Finally, the data show a negative association between fasting insulin concentrations and SMI% both in the PCOS group (r = −0.55) and in the control group (r = −0.55) after adjustment for age while the Matsuda index was positively associated with SMI% in all groups after age adjustment (*p* ≤ 0.05). The authors conclude by pointing out that women with PCOS show early signs of OS probably due to a disruption of insulin function.

### 3.5. Osteosarcopenia and Cardiovascular Diseases

In the scientific literature, the prevalence of sarcopenia, osteopenia, and osteoporosis in cardiac patients is widely discussed, while there are very few works devoted to the combination of these conditions. Bazdyrev and colleagues [63] investigated the prevalence of musculoskeletal disorders in 387 patients aged 50–82 years with stable coronary artery disease (CAD). The most common type of musculoskeletal disorder was sarcopenia with 13.4%; osteopenia/osteoporosis was detected in 7.2%, and; OS in 6.5%. The most pronounced clinical manifestation is reflected by a higher score on the SARCF questionnaire (screening for sarcopenia), low handgrip strength, small area of muscle tissue, low musculoskeletal index, as well as low values of BMD, were seen in patients with OS. Fahimfar and colleagues [64] assessed the link between OS and cardiovascular disease risk factors (such as age, education, smoking, physical activity, BMI, hypertension, diabetes, dyslipidemia ad high-fat mass) in 2426 participants aged ≥60 years. The results show a prevalence of OS of 34% and, in both genders, increased by age, while the slope was higher in men with an increase from 14.3% in men aged 60–64 years to 59.4% in men aged ≥75 years and an increase from 20.3% in women aged 60–64 years to 48.3% in women aged ≥75 years (*p* = 0.019). Furthermore, BMI was inversely associated with OS, more likely in individuals with high-fat mass. So, the protective role of greater BMI levels in association with the harmful effect of high-fat mass highlights the importance of lean muscle mass. OS was more likely in diabetic men (adjusted prevalence ratio: 1.33, 95% CI 1.04–1.69), but not in women. While no association between OS and smoking and lipid profiles has been found.

### 3.6. Osteosarcopenia and Anemia

Anemia (the pathological reduction of hemoglobin below normal levels) occurs often in older persons, and it is also considered a risk factor for osteoporosis [65], sarcopenia [66], falls, and fractures [67,68]. Hassan and colleagues [69] aimed to compare hemoglobin (Hb) levels in osteosarcopenic older subjects versus those with sarcopenia, osteopenia/osteoporosis alone, and controls. The study involved 558 community-dwelling participants older than 65 years. Anemia prevalence was 31.5% and it was highest among sarcopenic patients (39%), followed by osteosarcopenic (34%), osteoporotic/penic (26%), and controls (24%). Osteosarcopenic patients on average had 6.3 g/L lower Hb levels compared to controls (*p* = 0.001), and 3.7 g/L lower Hb than patients with osteoporosis/penia (*p* < 0.026). Interestingly, levels of Hb did not differ between sarcopenic vs. osteosarcopenic patients (*p* = 0.817) and between osteoporotic/osteopenic patients vs. controls (*p* > 0.259). The authors concluded that sarcopenia and OS (but not osteoporosis alone) are associated with anemia but further research is needed to confirm the results in another population and to explore possible biological mechanisms involved.

**Table 1 ijms-23-05591-t001:** Relevant results in main clinical studies on Osteosarcopenia (OS).

Author, Year [Ref]	Sample Size Age and Sex	% of OS	Relevant Results
Huo, 2015 [39]	680 (65% W) mean age 79 yrs	38%	OS subjects were older, mostly women, with a body mass index (BMI) below 25 and at higher risk of depression and malnutrition
Reiss, 2019 [40]	141 (60% W) 80.6 ± 5.5 yrs	14.2%	BMI and Mini Nutritional Assessment-Short form were lower in OS compared to sarcopenia or osteoporosis alone (*p* < 0.05)
Okayama, 2022 [41]	61 W 77.6 ± 8.1 yrs	39.3%	Patients with OS had lower quality of life scores, greater postural instability, and a higher incidence of falls.
Park, 2021 [42]	885 (67.1% W) 70.3 ± 6.2 yrs	19.2%	Disability (17.5, 95% CI: 14.8–20.1), frailty (3.0, 95% CI: 2.6–3.4), and depression mean score (4.6, 95% CI: 3.9–5.4) were statistically significantly higher in the OS group compared the other groups.
Pourhassan, 2021 [43]	572 (78% W) 75.1 ± 10.8 yrs	8%	OS patients were older and frailer and had lower BMI, fat, muscle mass, handgrip strength, and T-score compared to non-OS patients.
Salech, 2020 [44]	1119 (68.6% W) 72.0± 6.7 yrs	16.4%	OS increases with age from 8.9% at 60–69.9 years), to 33.7% (>80 years) (*p* < 0.0001; mortality was significantly higher for the group with OS (15.9%) compared with those without the condition (6.1%). The risk of falls, fractures and functional impairment increases in OS (falls: HR 1.60; CI 1.07–2.38; *p* < 0.05; fractures HR 1.54; CI 1.13–2.08; *p* < 0.01; functional impairment: HR 1.83; CI 1.41–2.38; *p* < 0.001).
Fahimfar, 2022 [45]	341 M 73.3 ± 7.4 yrs	100%	Risk of falls: positively associated with age (OR = 1.09, 95% CI: 1.04–1.14), fasting blood glucose, an increase of 10 mg/dL increased the chance of falling by 14% (OR = 1.14, 95% CI = 1.06–1.23); negatively associated with triglyceridemia (OR = 0.33, CI 95% = 0.12 to 0.89).
Di Monaco, 2020 [46]	350 W^2^ 79.7 ±7.2 yrs	65.7%	Significant difference in Spine Deformity Index (SDI) scores across the 3 groups (no osteoporosis and sarcopenia; osteoporosis or sarcopenia and osteosarcopenia (*p* < 0.001).
Drey, 2016 [48]	68 pre-frail older (47 W, 21 M) 65–94 yrs	41%	OS participants showed significantly reduced hand-grip increased chair rising time, and STS power time as well as significantly increased bone turnover markers.
Saeki, 2020 [49]	291 (137 M 154 W) 59–76 yrs	16.8%	OS and vertebral fracture were often seen in patients with frailty than in those without frailty (48.1% vs. 4.8% and 49.4% vs. 18.1%, respectively; *p* < 0.001). Frailty was an independent factor associated with OS (OR= 9.837; *p* < 0.001), and vice versa (OR = 10.069; *p* < 0.001).
Inoue, 2021 [50]	495 (68.7% W) 76.5 ± 7.2 yrs	11.1%	Logistic regression analysis revealed that OS was significantly associated with social frailty (pooled OR: 2.117; 95%CI: 1.104–4.213)
Inoue, 2022 [51]	432 patients (298 W) 75.9 ± 7.3 yrs	10.2%	Logistic regression analysis revealed that OS was independently associated with cognitive frailty with a higher odds ratio (OR: 8.246, 95% CI 3.319−20.487) than osteoporosis or sarcopenia alone.
Liu, 2021 [54]	150 (80 M and 70 W) patients with T2DM aged ≥50 yrs.	29%	Patients with OS had lower body mass index, waist circumference, body fat percentage (*p* < 0.001), AUC-Ins/Glu (*p* = 0.01), and AUC-CP/Glu (*p* = 0.013). Both AUC-Ins/Glu (OR = 0.634, *p* = 0.008) and AUC-CP/Glu (OR = 0.491, *p* = 0.009) were negatively associated with the presence of OS.
Pechmann, 2021 [55]	T2DM group *n* = 177, (64.4% W) 65.1 8.2 yrs; Control group *n* = 146, (54.7% W) 68.8 ± 11.0 yrs	11.9% (T2DM group); 2.14% (control group)	T2DM group versus the control group had higher rates fractures (29.9% vs. 18.5%, respectively, *p*= 0.019), lower handgrip strength values (24.4 ± 10.3 kg vs. 30.9 ± 9.15 kg, respectively, *p* < 0.001), but comparable BMD values. OS was associated with diabetes complications (*p* = 0.03), calcium and vitamin D supplementation (*p* = 0.01), and all components of OS diagnosis (*p* < 0.05).
Park, 2018 [56]	1344 Post-menopausal W >50 yrs	24.1%	Pro-inflammatory diet was associate with increased odds for osteopenic obesity (OR = 2.757, 95% CI: 1.398–5.438, *p* < 0.01) and OS obesity (OR = 2.186, 95% CI: 1.182–4.044, *p* < 0.05) respectivelyA deficiency of antioxidant vitamins (A and E) was found in OS subjects compared to control (*p* < 0.001).
Bazdyrev, 2021 [63]	387 stable coronary artery disease (26.9% W) 50–82 yrs	6.5%	Patients with OS had a higher score on the SARC-F questionnaire, low handgrip strength, small area of muscle tissue, low musculoskeletal index, as well as low values of bone mineral density.
Fahimfar, 2020 [64]	2353 (51.2% W) >60 yrs	34%	OS increases with age (from 14.3% in aged 60–64 years to 59.4% in aged ≥75 years in men and from 20.3% in aged 60–64 years to 48.3% in aged ≥75 years in women- *p* = 0.019). BMI was inversely associated with OS. High-fat mass was positively associated with OS [PR 1.46 (95% CI 1.11–1.92) in men, and 2.25 (95% CI 1.71–2.95) in women]. OS was more likely in diabetic men (adjusted PR: 1.33, 95% CI 1.04–1.69), but not in women. No association between OS and smoking and lipid profiles has been found.
Hassan, 2020 [69]	558 community-dwelling participants older (79 ± 7.5 yrs)	36%	OS patients on average had 6.3 g/L lower Hb levels compared to controls (*p* = 0.001), and 3.7 g/L lower Hb than patients with osteoporosis/penia (*p* < 0.026). Sarcopenia and OS (but not osteoporosis alone) are associated with anemia

W = women; M = men; T2DM = Type 2 diabetes mellitus; STS = Sit-to stand test.

## 4. Management of Osteosarcopenia

Management of osteosarcopenia currently relies on treatments used for the individual components of the disease, bone loss, and muscle loss, along with nutritional supplementation and exercise. The intervention studies are based on specific exercises with various levels of intensity and duration over time, often associated with the achievement of adequate levels of protein, calcium, and vitamin D through diets or supplements or the administration of pharmacological treatments. 

### 4.1. Non-Pharmacological Interventions

Multidisciplinary programs have been proposed to prevent and/or treat OS and reduce the risk of falls and frailty. Choi and colleagues [70] in a study conducted on 7000 subjects over the age of fifty years investigated the association between dietary calcium and phosphorus intake and changes in bone, muscle, and fat mass related to body composition. The results highlight a significant negative relationship between calcium and phosphorus intake with OS. These data add further knowledge on the management of OS, but further studies will have to confirm these results and thus improve the management of OS and its related components.

Gomez and colleagues [71] proposed a joint model of care for the assessment and prevention of osteoporosis and falls in an outpatient setting. The Falls and Fractures Clinic (FFC), consisting of a team of experts, includes a range of interventions: Vitamin D and calcium supplementation, osteoporosis medication, adjustment to current treatment regimens, supervised group exercise programs, proprioceptive/vestibular retraining exercises, physical therapy (gait and balance training), protein supplementation, hip protection, occupational therapy, and referrals to other specialists. Their multidisciplinary FFC reduced the risk of falls and fractures in older people at high risk of these adverse events, even over a relatively short time of 6 months. The results highlight the multidisciplinary approach to the management and prevention of OS and its components in the elderly. 

Exercise is an eligible treatment for OS, but which exercise is right? In their review, Hong and Kim [72] investigate this question. Physical training is recommended as a low-cost and safe non-drug intervention strategy for keeping musculoskeletal health [73]. The specific mechanisms by which exercise improves bone health are not yet fully understood. However, it is widely accepted that the mechanical load induced by physical training increases muscle mass, produces mechanical stress in the skeleton, and increases the activity of osteoblasts [74,75]. However, not all exercise modalities are equally osteogenic. Physical training with an osteogenic effect must have a mechanical load applied to the bones higher than that during daily activities. Weighted impact exercises such as jumping and/or progressive resistance exercise (RE), alone or in combination, can improve bone health in adults [73]. Hong and Kim [72] review the effects of RE on musculoskeletal health, particularly on bone strength. RE is a physical conditioning program that improves fitness, health, and sports performance, with different training modes (free weights, weight machines, medicine balls, rubber bands, and different movement speeds). Mechanical loading is a fundamental factor for bone mass growth. The principle of loading was first developed by Frost [76]. This system involves the osteocytes that produce a protein, Sclerostin, which plays a central role in the regulation and formation of bone [77]. RE, either alone or in combination with other interventions, may be the most optimal strategy to improve the muscle and bone mass in postmenopausal women, middle-aged men, or the older population.

An important study on OS management is represented by the FrOST Study (Franconian Osteopenia and Sarcopenia Trial) [78]. The project aimed to evaluate the effect of high-intensity dynamic resistance exercise (HIT-DRT) and whey protein supplementation (WPS) on BMD and sarcopenia parameters in osteosarcopenic men. Additionally, cholecalciferol and calcium supplementation are considered. The study was conducted between February 2018 and February 2020 on a final sample of 43 osteosarcopenic community-dwelling men (73–91 years). The trial was applied for 18 months and volunteers were randomly assigned to either an active group (HIT-DRT *n* = 21) or a control group (CG *n* = 22) that maintained their habitual lifestyle. Both received dietary protein (up to 1.5 g/kg/day in HIT-DRT and 1.2 g/kg/day in CG) and vitamin D supplements (up to 800 IE/day). The detailed procedures such as intermedia (6, 8, 12, and 16 months) and final results are reported in several publications [78,79,80]. At six months from the start, the results showed a significant effect of the exercise intervention on the Z-score of sarcopenia in the HIT-DRT group (*p* < 0.001) and a significant worsening of the same in the CG (*p* = 0.012). This shows that without physical activity stimulation, sarcopenia worsens naturally, and the amount of protein supplement in the CG (1.2 g/kg/day) alone was ineffective in keeping muscle mass and function while HIT-DRT in combination with protein supplementation is a favorable intervention strategy to reduce the risks, progression, and burden of sarcopenia [78]. At 12-months, the authors report that the findings related to bone tissue and skeletal muscle mass index (SMI). The BMD mg/cm^3^) at the integral lumbar spine (LS) was maintained in the HIT-DRT group (0 ± 1.9%; *p* = 0.857) and decreased significantly in the CG (−1.2 ± 1.8%; *p* = 0.004). Additionally, for SMI (kg/m^2^) there is a significant increase in the HIT-DRT (3.6 ± 3.0%; *p* < 0.001) and significant decrease (−1.2 ± 1.9%; *p* = 0.03) in the CG. Moreover, the maximum dynamic strength of the hip and leg extensors at baseline and after 8 and 12 months increased significantly (*p* < 0.001) by 27 ± 15% in the HIT-DRT and remained the same (−1.4 ± 8.9%) in the CG (*p* = 0.599). Changes between groups are significant for all findings (*p* < 0.001). At 18 months, significant positive effects for sarcopenia Z-score (standardized mean difference (SMD): 1.40), BMD at the lumbar spine (SMD: 0.72) and total hip (SMD: 0.72) are reported. Furthermore, there are evident positive effects for the skeletal muscle mass (1.97, *p* < 0.001), while only moderate effects for functional sarcopenia parameters (0.87, *p* = 0.008; handgrip strength) and low positive effects for gain velocity (0.39, *p* = 0.209; gait velocity). The authors conclude by evaluating the application of the HIT-DRT protocol combined with moderate protein supplementation, conditioning exercises, and supervision by a personal trainer, it is effective in combating OS in older men with sarcopenia e osteoporosis [79]. Due to force majeure during the lockdown for COVID-19, there was a change in the protocol study interrupting the training period while the vitamin D supplementation continued. This change led to studying detraining and what happened during the forced break. The same authors [80] evaluated the effect of a 6-month period of de-training on muscle quality. During detraining, the HIT-DRT group lost approximately one-third of the HIT-DRT-induced gain and showed a significantly (*p* = 0.001) higher reduction in muscle quality than the CG. The negative effect was only significant for skeletal muscle mass index and hip/leg extensor strength (*p* = 0.002 and *p* = 0.013), but not for lumbar spine BMD (*p* = 0.068), total hip BMD (*p* = 0.069), handgrip strength (*p* = 0.066) or gait velocity (*p* = 0.067) for lean body mass (*p* = 0.001), total body fat (*p* = 0.003), and the MetS Z-score (*p* = 0.003) [79,80].Thus, exercise protocols for older people should focus on continuous exercise with short regeneration periods rather than on intermitted protocols with pronounced training breaks. Table 2 summarizes the relevant results on OS subjects in Non-Pharmacological Interventions.

### 4.2. Pharmacological Interventions

There are different pharmacological treatments to counteract the loss of bone mass [81]. The drug treatments approved by the FDA (Food and Drug Administration) include anti-resorption (denosumab, bisphosphonates), anabolic (teriparatide, abaloparatide), anti-sclerostin (romosozumab), and hormonal (hormone replacement therapy, selective estrogen receptor modulators). However, there are no specific treatments for OS. Two clinical studies have found benefits of denosumab on muscle and bone mass and muscle strength and balance in older people at risk of falls and fractures [82,83]. In this regard, further double-blind trials are needed to confirm the efficacy of denosumab in treating OS. Kleine [84], in a review on drug treatments for OS, reports that the use of antiresorptives for osteoporosis and the efficacy of bisphosphonates to prevent hydroxyapatite dissolution in treating postmenopausal osteoporosis have been known for several years. Furthermore, new generations of bisphosphonates have been increasingly more effective in preventing further bone loss and reducing fracture incidence in osteoporosis. Nevertheless, those drugs are not routinely used to prevent progression to osteoporosis in osteopenic patients. This drug is limited to those with more advanced stages of bone loss as diagnosed by osteoporosis. Drug treatment of OS, besides antiresorptive agents, consists of inhibitors of angiotensin-converting enzymes (ACE). New anti-myostatin antibodies are promising experimentally but have yet to undergo successful clinical trials. Ghrelin agonists can determine the improvement of malnutrition and increase muscle mass or strength [85]. While these latter agents may also improve OS, more studies are necessary to figure out their real value. Some potential elements common to the muscle and bone have been highlighted as targets for new therapies, including RANK L inhibition (exclusively antiresorptive action) or IGF-1. Denosumab, a monoclonal antibody used to treat osteoporosis to slow bone breakdown (reducing osteoclastogenesis) can block the receptor activator of nuclear factor-kappa-β ligand (RANK-L)) [86]. The positive potential of this drug was evaluated in the FREEDOM study [87], where an increase in BMD was seen in all bone sites with a decrease in fractures in postmenopausal osteoporotic women. This drug acting in the RANK L, which has also been associated with the atrophy-induced denervation process and muscle dysfunction, affects the muscle by increasing the ALM and muscle strength (handgrip) in the treated sample. On the other hand, IGF-1, together with GH, promotes proper bone growth and muscle mass production. The elderly have reduced muscle sensitivity to GH with reduced muscle mass and increased adipose tissue. Therapies based on the association of GH/IGF-1 have shown a link with the development of positive muscle mass. However, a positive association with a bone mass gain was not confirmed from a systematic review with meta-analysis where the evidence only confirmed a reduced fracture risk in women, with no increase in bone mass. However, the evidence to support these pharmacological treatments is still limited, given the lack of randomized controlled trials (RCT) in humans [88]. 

## 5. Conclusions

Osteosarcopenia is a complex syndrome that, if not recognized in time, leads to falls, fractures, loss of self-sufficiency in daily activities, and premature death. OS diagnosis has involved the determination of low muscle and bone mass along with evaluating muscle strength and physical performance. In these terms, diagnosis is based on the reference definitions of osteoporosis and sarcopenia and it is important to standardize these definitions to improve the diagnosis of OS in clinical practice. Biochemical assessment of bone and muscle metabolism has been proposed to improve the early diagnosis, screening, and assessment of the response to therapies in people with OS, but at the moment, it seems to play a secondary role and further research is needed in this field. There are not many clinical studies reporting results on subjects diagnosed with OS and this may be related to the difficulty of accurately diagnosing OS. Studies show the relationship between OS and risk of frailty, falls, and fractures and some association with NCDs pathologies such as diabetes, obesity, and cardiovascular disease, and, in some cases, the importance of deepening the related mechanisms are emphasized. Lifestyle changes to avoid factors such as sedentarism, obesity, and poor nutrition can fight this pathology. Practicing physical exercise combining resistance and balance with adequate nutrition and nutritional supplementations such as proteins, Vitamin D, or calcium, represents a significant strategy to brake OS. Additionally, pharmacological interventions may confer benefits on muscle and bone health. However, both nonpharmacological and pharmacological interventions require additional randomized controlled trials (RCT) in humans to deepen the synergistic effect of exercise, nutritional interventions, and drug compounds in osteosarcopenia.

## Figures and Tables

**Figure 1 ijms-23-05591-f001:**
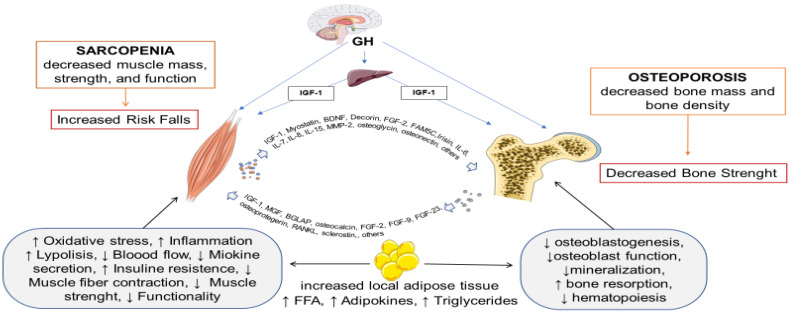
Pathogenesis and etiology of osteosarcopenia.

**Table 2 ijms-23-05591-t002:** Relevant results on Osteosarcopenia (OS) subjects in Non-Pharmacological Interventions.

Author, Year [Ref]	Sample Size Age and Sex	Type of Intervention	% of OS	Relevant Results
Gomez, 2018 [71]	106 (68% W) 78 ± 8 yrs	Multifactorial interventions: e.g., vitamin D/calcium supplement, osteoporosis medications, supervised group exercise programs; protein supplement, etc	53%	At 6-month follow-up, the multidisciplinary interventions reduce falls by more than 80% and 50% fracture risk. In addition, 65% of patients had a reduced risk for falling and a 57% reduction in 10-year fracture probability.
Lichtenberg, 2019 [78]	43 M (21 EG group; 22 Inactive Control group CG 73 to 91 yrs.	FROST Study 18 months trial High-intensity dynamic resistance exercise (HIT-DRT), whey protein supplement (up to 1.5 g/kg/day in HIT-DRT and 1.2 g/kg/day in CG); vitamin D supplements (up to 800 IE/day).	100%	The results show a significant effect of the exercise intervention on the sarcopenia Z-score (*p* < 0.001), a significant increase in the skeletal muscle mass index (SMI) (*p* < 0.001), and in handgrip strength (*p* < 0.001) in the HI-RT group and a significant worsening on the sarcopenia Z score in the CG group (*p* = 0.012).
Kemmler, 2020 [79]	43 M (21 EG group; 22 Inactive Control group CG 73 to 91 yrs.	FROST Study 18 months high-intensity dynamic resistance exercise (HIT-DRT), whey protein supplement (up to 1.5 g/kg/day in HIT-DRT and 1.2 g/kg/day in CG); vitamin D supplements (up to 800 IE/day).	100%	After 12 months the lumbar spine (LS) BMD was maintained in the EG and decreased significantly in the CG (*p* < 0.001; standardized mean difference (SMD) = 0.90); SMI increased significantly in the EG and decreased significantly in the CG (*p* < 0.001; SMD = 1.95). Changes in maximum hip−/leg extensor strength were much more prominent (*p* < 0.001; SMD = 1.92) in the EG.
Kemmler, 2021 [80]	43 M (21 EG group; 22 Inactive Control group CG 73 to 91 yrs.	FROST Study 6 months of detraining after 18 months of intervention.	100%	During detraining, the EG group lost approximately one-third of the HIT-DRT-induced gain and showed a significantly (*p* = 0.001) higher reduction in muscle quality than the CG. The negative effect was only significant for skeletal muscle mass index and hip/leg extensor strength (*p* = 0.002 and *p* = 0.013), but not for lumbar spine BMD (*p* = 0.068), total hip BMD (*p* = 0.069), handgrip strength (*p* = 0.066) and gait velocity (*p* = 0.067).

W = women; M = men; BMD = Bone Mineral Density.

## Data Availability

Not applicable.

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
