# Peer review of "Osteosarcopenia: A Narrative Review on Clinical Studies"

_ijms, 2022, doi:10.3390/ijms23105591_

Round 1

Reviewer 1 Report

This is a well written, interesting review article that will contribute to the literature.

Author Response

We would like to thank the reviewer for the effort they put into reviewing the manuscript and for the positive appreciation of our work.

Reviewer 2 Report

The manuscript entitled "Osteosarcopenia: a narrative review on clinical studies" by Angela Polito and colleagues presents a comprehensive and extremely exhaustive overview of the most relevant clinical trial focused on the management of osteosarcopenia.

 The reviewer has only one comment to this complete review which is the addition of summary tables.

Author Response

We would like to thank the reviewer for the effort they put into reviewing the manuscript and for the positive appreciation of our work. We have made corrections and replies which we hope meet with approval.

Our response to the reviewer’s advice is as follows:

Point 1: The reviewer has only one comment to this complete review which is the addition of summary tables.

Thank you for your suggestion. We have added two tables to the manuscript.

In table 1 we summarizes the relevant results in main clinical studies on OS (changes are highlighted in red at page 5 and page 8 in the revised manuscript).

In table 2 we summarizes the relevant results on OS subjects in Non-Pharmacological Interventions. (changes are highlighted in red at page 13 in the revised manuscript).

Table 1. Relevant results in main clinical studies on Osteosarcopenia (OS)

Author, Year [ref]

Sample size

Age and sex

% of OS

Relevant results

Huo, 2015 [39]

680

(65%W)

mean age  79 yrs

38%

OS subjects were older, mostly women, with a body mass index (BMI) below 25 and at higher risk of depression and malnutrition

Reiss, 2019 [40]

141

(60% W)

80.6 ± 5.5 yrs

14.2%

BMI and Mini Nutritional Assessment- Short form were lower in OS compared to sarcopenia or osteoporosis alone (p < 0.05)

Okayama, 2022 [41]

61 W

77.6 ± 8.1 yrs

39.3%

Patients with OS had lower quality of life scores, greater postural instability, and a higher incidence of falls.

Park, 2021 [42]

885

(67.1% W)

70.3 ± 6.2 yrs

19.2%

Disability (17.5, 95% CI: 14.8- 20.1), frailty (3.0, 95% CI: 2.6-3.4), and depression mean score (4.6, 95% CI: 3.9-5.4) were statistically significantly higher in the OS group compared the other groups.

Pourhassan, 2021 [43]

572

(78% W)

75.1±10.8 yrs

8%

OS patients were older and frailer and had lower BMI, fat, and muscle mass, handgrip strength, and T-score compared to non OS patients.

Salech, 2020 [44]

1119

(68.6% W)

72.0± 6.7 yrs

16.4%

OS increases with age from 8.9% at 60-69.9 years), to 33.7% (> 80 years) (P < .0001; mortality was significantly higher for the group with OS (15.9%) compared with those without the condition (6.1%). The risk of falls, fractures and functional impairment increases in OS (falls: HR 1.60; CI 1.07-2.38; P<0.05; fractures HR 1.54; CI 1.13-2.08; P < 0.01; functional impairment: HR 1.83; CI 1.41-2.38; P<0.001).

Fahimfar, 2022  [45]

341 M

73.3±7.4 yrs

100%

Risk of falls: positively associated with age (OR=1.09, 95% CI: 1.04–1.14), fasting blood glucose, an increase of 10 mg/dl increased the chance of falling by 14% (OR = 1.14, 95% CI = 1.06–1.23); negatively associated with triglyceridemia (OR=0.33, CI 95%=0.12 to 0.89).

Di Monaco, 2020 [46]

350 W2

79.7 ±7.2 yrs

65,7%

Significant difference in Spine Deformity Index (SDI) scores across the 3 groups (no osteoporosis and sarcopenia; osteoporosis or sarcopenia and osteosarcopenia (P <0.001).

Drey, 2016 [48]

68 pre-frail older

(47 W, 21 M)

65-94 yrs

41%

OS participants showed significantly reduced hand grip increased chair rising time, and STS power time as well as significantly increased bone turnover markers.

Saeki, 2020 [49]

291

(137M 154W)

59-76 yrs

16,8%

OS and vertebral fracture were often seen in patients with frailty than in those without frailty (48.1% vs. 4.8% and 49.4% vs. 18.1%, respectively; P< 0.001). Frailty was an independent factor associated with OS (OR= 9.837; p < 0.001), and vice versa (OR=10.069; p < 0.001).

Inoue, 2021 [50]

495

(68.7% W)

76.5 ± 7.2 yrs

11,1%

 Logistic regression analysis revealed that OS was significantly associated with social frailty (pooled OR: 2.117; 95%CI: 1.104–4.213)

Inoue, 2022 [51]

432 patients

(298W)

75.9 ±7.3 yrs

10.2%

Logistic regression analysis revealed that OS was independently associated with cognitive frailty with a higher odds ratio (OR: 8.246, 95% CI 3.319− 20.487) than osteoporosis or sarcopenia alone.

Liu, 2021 [54]

150 (80 M and 70 W) patients with T2DM aged ≥50 yrs.

29%

Patients with OS had lower body mass index, waist circumference, body fat percentage (p <0.001), AUC-Ins/Glu (P=0.01), and AUC-CP/Glu (P=0.013). Both AUC-Ins/Glu (OR = 0.634, P = 0.008) and AUC-CP/Glu (OR = 0.491, P = 0.009) were negatively associated with the presence of OS.

Pechmann, 2021 [55]

T2DM group n=177, (64.4%W)

65.1 8.2 yrs;

Control group n=146, (54.7%W)

68.8±11.0 yrs

11.9% (T2DM group);

 2.14% (control group)

T2DM group versus the control group had higher rates fractures (29.9% vs 18.5%, respectively, p= 0.019), lower handgrip strength values (24.4 ± 10.3 kg vs 30.9 ± 9.15 kg, respectively, P< 0.001), but comparable BMD values.

OS was associated with diabetes complications (P= 0.03), calcium and vitamin D supplementation (P= 0.01), and all components of OS diagnosis (P< 0.05).

Park, 2018 [56]

1344

Post-menopausal W

>50 yrs

24.1%

Pro-inflammatory diet was associate with increased odds for osteopenic obesity (OR = 2.757, 95% CI: 1.398-5.438, p<0.01) and OS obesity (OR = 2.186, 95% CI: 1.182-4.044, p<0.05) respectively

A deficiency of antioxidant vitamins (A and E) was found in OS subjects compared to control (p<0.001).

Bazdyrev, 2021 [63]

387 stable coronary artery disease

(26.9 % W)

50-82 yrs

6.5%

Patients with OS had higher score on the SARC-F questionnaire, low handgrip strength, small area of muscle tissue, low musculoskeletal index, as well as low values of bone mineral density. 

Fahimfar, 2020 [64]

2353

(51.2% W)

>60 yrs

34%

OS increases with age (from 14.3% in aged 60–64 years to 59.4% in aged ≥ 75 years in men and from 20.3% in aged 60–64 years to 48.3% in aged ≥ 75 years in women- p= 0.019). BMI was inversely associated with OS.  High-fat mass was positively associated with OS [PR 1.46 (95% CI 1.11-1.92) in men, and 2.25 (95% CI 1.71-2.95) in women]. OS was more likely in diabetic men (adjusted PR: 1.33, 95% CI 1.04–1.69), but not in women. No association between OS and smoking and lipid profiles has been found.

Hassan, 2020 [69]

558 community-dwelling

participants older

(79 ± 7.5 yrs)

36%

OS patients on average had 6.3 g/L lower Hb levels compared to controls (p=0.001), and 3.7 g/L lower Hb than patients with osteoporosis/penia (p<0.026). Sarcopenia and OS (but not osteoporosis alone) are associated with anemia

W= women; M= men; T2DM= Type 2 diabetes mellitus; STS= Sit-to stand test

Table 2. Relevant results on Osteosarcopenia (OS) subjects in Non-Pharmacological Interventions

Author, Year [ref]

Sample size

Age and sex

Type of intervention

% of OS

Relevant results

Gomez, 2018 [71]

106

 (68%W)

78±8 yrs

Multifactorial interventions: e.g. vitamin D/calcium supplement, osteoporosis medications, supervised group exercise programmes; protein supplement etc

53%

At 6-month follow-up, the multidisciplinary  interventions reduce falls by more than 80% and 50% fracture risk. In addition, 65% of patients had a reduced risk for falling and a 57% reduction in 10-year fracture probability.

Lichtenberg, 2019 [78]

43 M

(21 EG group; 22 Inactive Control group CG9

73 to 91 yrs.

FROST Study 18 months trial

 High intensity dynamic resistance exercise (HIT-DRT), whey protein supplement (up to 1.5 g/kg/day in HIT-DRT and 1.2 g/kg/day in CG); vitamin D supplements (up to 800 IE/day).

100%

The results show a significant effect of the exercise intervention on the sarcopenia Z-score (p<0.001), a significant increase of skeletal muscle mass index (SMI) (p<0.001) and on handgrip strength (p<0.001) in the HI-RT group and a significant worsening on the sarcopenia Z score in the CG group (p=0.012).

Kemmler, 2020 [79]

43 M

(21 EG group; 22 Inactive Control group CG9

73 to 91 yrs.

FROST Study 18 months high intensity dynamic resistance exercise (HIT-DRT), whey protein supplement (up to 1.5 g/kg/day in HIT-DRT and 1.2 g/kg/day in CG); vitamin D supplements (up to 800 IE/day).

100%

After 12 months the lumbar spine (LS) BMD was maintained in the EG and decreased significantly in the CG (p < 0.001; standardized mean difference (SMD) = 0.90); SMI increased significantly in the EG and decreased significantly in the CG (p < 0.001; SMD = 1.95). Changes in maximum hip−/leg extensor strength were much more prominent (p < 0.001; SMD = 1.92) in the EG.

Kemmler, 2021 [80]

43 M

(21 EG group; 22 Inactive Control group CG9

73 to 91 yrs.

FROST Study 6 months of detraining after 18 months of intervention.

100%

During detraining, the EG group lost approximately one third of the HIT-DRT -induced gain and showed a significantly (p=0.001) higher reduction in muscle quality than the CG. The negative effect was only significant for skeletal muscle mass index and hip/leg extensor strength (p=0.002 and p=0.013), but not for lumbar spine BMD (p=0.068), total hip BMD (p=0.069), hand grip strength (p=0.066) and gait velocity (p=0.067).

W= women;  M= men;  BMD= Bone Mineral Density
